# Risk Factors for the Impairment of Ambulation in Older People Hospitalized with COVID-19: A Retrospective Cohort Study

**DOI:** 10.3390/ijerph20227057

**Published:** 2023-11-13

**Authors:** Erika Christina Gouveia e Silva, Ana Carolina Basso Schmitt, Caroline Gil de Godoy, Danielle Brancolini de Oliveira, Clarice Tanaka, Carlos Toufen, Carlos Roberto Ribeiro de Carvalho, Celso R. F. Carvalho, Carolina Fu, Keith D. Hill, José Eduardo Pompeu

**Affiliations:** 1Department of Physical Therapy, Speech and Occupational Therapy, School of Medicine—University of Sao Paulo, Brazil. R. Cipotanea, 51-Vila Butanta, São Paulo 05360-160, Brazil; carolinaschmitt@usp.br (A.C.B.S.); caroline.gil@usp.br (C.G.d.G.); danibrancolini@usp.br (D.B.d.O.); cltanaka@usp.br (C.T.); cscarval@usp.br (C.R.F.C.); carolfu@usp.br (C.F.); j.e.pompeu@usp.br (J.E.P.); 2Division of Pulmonology, Heart Institute (InCor), School of Medicine—University of Sao Paulo, Brazil. Av. Dr. Eneas Carvalho de Aguiar, 44-Cerqueira Cesar, São Paulo 05403-900, Brazilcarlos.carvalho@hc.fm.usp.br (C.R.R.d.C.); 3Rehabilitation Ageing and Independent Living (RAIL) Research Centre, Monash University, Melbourne, VIC 3199, Australia; keith.hill@monash.edu

**Keywords:** COVID-19, functional status, hospitalization, older people

## Abstract

(1) Background: Some older people hospitalized with COVID-19 have experienced reduced ambulation capacity. However, the prevalence of the impairment of ambulation capacity still needs to be established. Objective: To estimate the prevalence of, and identify the risk factors associated with, the impairment of ambulation capacity at the point of hospital discharge for older people with COVID-19. (2) Methods: A retrospective cohort study. Included are those with an age > 60 years, of either sex, hospitalized due to COVID-19. Clinical data was collected from patients’ medical records. Ambulation capacity prior to COVID-19 infection was assessed through the patients’ reports from their relatives. Multiple logistic regressions were performed to identify the risk factors associated with the impairment of ambulation at hospital discharge. (3) Results: Data for 429 older people hospitalized with COVID-19 were randomly collected from the medical records. Among the 56.4% who were discharged, 57.9% had reduced ambulation capacity. Factors associated with reduced ambulation capacity at discharge were a hospital stay longer than 20 days (Odds Ratio (OR): 3.5) and dependent ambulation capacity prior to COVID-19 (Odds Ratio (OR): 11.3). (4) Conclusion: More than half of the older people who survived following hospitalization due to COVID-19 had reduced ambulation capacity at hospital discharge. Impaired ambulation prior to the infection and a longer hospital stay were risks factors for reduced ambulation capacity.

## 1. Introduction

Older people are more vulnerable to the adverse effects of hospitalization [1], including the impairment of functional capacity [2]. The greatest vulnerability is associated with the decline of several bodily systems and lower functional reserves due to aging [3,4,5], in addition to the reduced response to stressful stimuli [3,4]. Therefore, older people are more susceptible to immobility, sepsis, and hypoxemia [6]. Thus, hospital admissions can impair various functions of older people, including ambulation capacity [2]. 

Worldwide, there have been more than 771,154,224 confirmed cases of COVID-19, and more than 6,960,783 deaths caused by the disease [7]. The highest prevalence of severe cases was found in older people hospitalized with previous comorbidities (such as obesity, arterial hypertension, and diabetes) [8,9,10,11,12,13]. Treatment of severe COVID-19 requires long periods of hospitalization and bed rest [14,15]. Several studies have shown the negative impact of long hospital stays and immobility on the functional capacity of older people [12,16,17,18,19]. Thus, the tissue damage in multiple systems directly caused by the cellular invasion of SARS-CoV-2 [9], the intense systemic inflammatory response [8], and the adverse effects of hospitalization can cause a greater functional impact on older people [11]. Muscle inactivity and intensive care interventions, including mechanical ventilation, neuromuscular blocking agents [20], and prone positioning [21] can decrease muscle strength, joint mobility, and respiratory capacity [20,21,22], especially in older people [19].

These structural and functional consequences in critically ill patients can persist even after hospital discharge, and are known as post-intensive care syndromes [8]. They can trigger impairments in the performance of daily activities, negatively affect functionality [23,24], autonomy, and independence, and increase the risk of mortality [25,26].

The longer the stay in the intensive care unit, the greater the risk of physical, cognitive, and emotional problems [27]. Approximately 25–45% of critically ill patients exhibit neuromuscular complications during and after intensive care, including symmetrical flaccid limb paralysis resulting from systemic inflammatory responses [28], which is associated with the use of steroid medication that can cause significant adverse effects, including immune dysfunction and sarcopenia [16].

Studies show that approximately 65% of older people hospitalized due to COVID-19 suffer a loss of mobility [16,19]. Still, the clinical and sociodemographic factors associated with functional decline and the prevalence of the impairment of ambulation capacity at hospital discharge have not yet been established.

Thus, the objectives of this study were to estimate the prevalence and identify the factors associated with the worsening of ambulation capacity at hospital discharge in older people admitted with COVID-19.

## 2. Materials and Methods

This retrospective study was performed at a referral public hospital for COVID-19 in Sao Paulo, Brazil, and was approved by the local ethics committee (number 4.052.246).

Data were collected from a random sample of the medical records and managed using Research Electronic Data Capture (REDCap), which is a secure environment to store highly sensitive information. The inclusion criteria were older people aged 60 years or more [29]; both sexes; and hospitalized with a diagnosis of COVID-19 during the period from 1 April 2020 to 30 November 2020 (corresponding to the 1st wave), or from 1 January 2021 to 31 May 2021 (2nd wave). Cases with missing or dubious information in the medical records were excluded.

Sociodemographic data (age, sex, and race) and habits (tobacco and alcohol use) were processed. The clinical data consisted of the current COVID-19 wave at the point of admission (1st or 2nd); hospital outcome (discharge or death); comorbidities (immunosuppression, hematological, neurological, pulmonary, cardiovascular, renal, hepatic, systemic arterial hypertension, diabetes mellitus, obesity, and dyslipidemia); length of hospital stay and intensive care unit stay; and use of mechanical ventilation. The main outcome was the ambulation capacity before COVID-19 and at the point of hospital discharge. Patients’ ambulation capacity is evaluated and registered in their medical records by the physiotherapy team as a routine. Especially for this study, ambulation capacity prior to COVID-19 infection was assessed through the patients’ medical records.

Ambulation capacity classifications were adapted from Mehrholz et al. [30]. In order to decrease the number of categories, we grouped the following classifications:i.Independent ambulator: Individual can ambulate independently on flat or uneven surfaces, stairs, and uneven slopes. This classification includes the categories “independent ambulator only on a flat surface” and “independent ambulator”.ii.Dependent ambulator: Individual requires manual contact with, at most, one person while ambulating on level surfaces in order to prevent falls; manual contact consists of continuous or intermittent touch to aid balance or coordination. This classification includes the categories “level III physical assistance-dependent ambulator” and “level I physical assistance-dependent ambulator and supervision-dependent ambulator”.iii.Nonfunctional ambulator: Requires maximum help, with the need for assistive technology.

Ambulation capacities at the point of hospital discharge were divided into two groups, according to their evolution: (a) “the same” (ambulation capacity did not change during hospitalization); and (b) “worse” (an independent ambulator prior to COVID-19 became a dependent or non-functional ambulator, or a dependent ambulator prior to COVID-19 became a non-functional ambulator at hospital discharge).

The majority of patients with moderate or severe COVID-19 received corticosteroids, supplementary oxygen, and anti-thrombotic drugs [31], as well as rehabilitation with an early mobilization protocol [32].

### Data Analysis

Statistical analysis was performed using Stata 16 (StataCorp, College Station, Texas, USA). Data distribution was analyzed through the Shapiro–Wilk statistic. Data did not follow a normal distribution, and thus, median and interquartile ranges were used for the continuous variables. Frequency (number and percentage) was presented for the nominal variables. The prevalence of the impairment of ambulation capacity at hospital discharge was estimated using a confidence interval. Sociodemographic and clinical characteristics were presented according to the ambulation capacity classification before COVID-19. The chi-squared test was used to verify the association between ambulation capacity before COVID-19 and in-hospital death, as well as the ambulation capacity evolution (the same or worse) at hospital discharge.

Bivariate logistic regression analysis was calculated for each independent variable with the impairment of ambulation capacity at hospital discharge. For the multivariate regression model, sociodemographic and clinical variables with *p* < 0.20 were included, and a correlation matrix was performed on one of the pair of highly correlated variables for the multivariate regression (length of hospital stay and length of intensive care unit stay). Finally, a multivariate regression model—adjusted for sex, race, tobacco use, obesity, dyslipidemia, and neurological diseases—was built to identify possible predictors for reduced ambulation capacity at hospital discharge. We adopted a significance level of 0.05.

## 3. Results

### 3.1. Older People Hospitalized with COVID-19

Data from 429 older people hospitalized with COVID-19 were randomly collected from the medical records. The mean age of the older people included in this study was 68 (63–74) years, most were male (60.1%, *n* = 258), 32.0% (*n* = 136) were brown, and 33.6% (*n* = 143) were black. Before COVID-19 infection, 67.4% (*n* = 289) of the older people were classified as independent ambulators, 15.8% (*n* = 68) as dependent ambulators, and 16.8% (*n* = 72) as non-functional ambulators. 

In-hospital death occurred in 187 patients (43.6%). Patients who died during hospitalization had reduced ambulation capacity prior to COVID-19 (OR: 2.6; CI95%: 1.7–4.9; *p* < 0.001).

Among the 242 (56.4%) who were discharged, 42.1% (*n* = 102) maintained the same level of ambulation as before COVID-19, and 57.9% (*n* = 140) suffered reduced ambulation capacity (Flowchart 1). Hence, the prevalence of the impairment of ambulation capacity at hospital discharge was 57.9% (CI95%: 51.4–64.1%).

Age, race, tobacco use, neurological disease, use of invasive mechanical ventilation, in-hospital death, and current pandemic wave had an association with the classification of ambulation capacity prior to COVID-19 (Table 1).

### 3.2. Ambulation Capacity at Hospital Discharge

Table 2 shows that 105 (24.5%) individuals classified as independent and 35 (8.2%) individuals classified as dependent ambulators prior to COVID-19 had reduced ambulation capacity at hospital discharge. There was a significant difference between the level of ambulation capacity before COVID-19 and at hospital discharge (χ^2^ = 50.69, df = 6, *p* < 0.001), as shown in Table 2 below. In-hospital deaths of independent ambulators, dependent ambulators, and non-functional ambulators were 36.7% (*n* = 106), 45.6% (*n* = 31), and 69.4% (*n* = 50), respectively. The proportions of reduced ambulation capacity for the independent ambulators and dependent ambulators were 36.3% (*n* = 105) and 51.5% (*n* = 35), respectively. Non-functional ambulators did not experience any change in their ambulation capacity at hospital discharge.

Table 3 shows the demographic and clinical characteristics of the older people who were discharged from the hospital *(n* = 242). The median age of the survivors was 67 years (63–73 interquartile range), and 52.9% (*n* = 128) were male. Most of the older people (57.9%, 95%CI: 51.4–64.1%) had a reduced ambulation capacity at hospital discharge. Table 3 also presents the distribution and bivariate regression of demographic and clinical characteristics related to the ambulation capacity change at hospital discharge. The factors associated with reduced ambulation capacity were male sex; brown or black race; neurological disease; dyslipidemia; obesity; length of hospital stay longer than 20 days; intensive care stay longer than 11 days; being a dependent or non-functional ambulator prior to COVID-19; and hospitalization during the 2nd wave. 

Table 4 shows the predictors for reduced ambulation capacity at hospital discharge. The worsening of ambulation capacity was independent of sex, race, tobacco use, obesity, dyslipidemia, and neurological diseases, but was associated with hospital stays longer than 20 days (OR: 3.5; CI95%: 1.7–7.3; *p* = 0.001), being a dependent ambulator before COVID-19 (OR: 11.3; CI95%: 1.4–52.7; *p* = 0.002), and hospitalization during the 2nd wave (OR: 4.8; CI95%: 2.1–11.1; *p* < 0.001).

## 4. Discussion

The prevalence of the decreased ambulation capacity of older adults hospitalized due to severe COVID-19 was 57.9%. The factors associated with the worsening in ambulation capacity at hospital discharge were hospital stays higher than 20 days; decreased ambulation capacity before COVID-19; and hospitalization during the 2nd wave of the pandemic.

Our findings concerning the worsening in ambulation capacity are consistent with studies in which acute post-COVID-19 patients presented with alterations in musculoskeletal and cardiorespiratory function [27,33]. Welch et al. [14] point out that a decline in muscle trophism and function may be common in patients with COVID-19. However, our in-hospital mortality (43.6%) was higher than previously reported (20% to 31%) [8,19,34]. This finding may be related to our data collection site, which was a referral hospital for severe and moderate cases of COVID-19, where only patients with significant clinical worsening were transferred; these patients often required mechanical ventilation.

Our results showed that older adults with a hospital stay longer than 20 days presented a higher chance of suffering worsening in their ambulation capacity. In our study, the older adults stayed longer in the hospital (median 17 days) than in other studies (with a median hospital stay of 12 days) [10,13]. In a study that compared muscle strength and mobility between COVID-19 and non-COVID-19 patients, the results showed that muscle strength and mobility at ICU discharge are associated with the length of stay during COVID-19 infection [35]. Indeed, higher serum concentrations of inflammatory cytokines are seen in patients with COVID-19 who require intensive care [14,36] and stay longer in the hospital. This has negative consequences on muscle protein synthesis, resulting in a state of anabolic resistance, which requires a higher protein intake to stimulate muscle protein synthesis [14,27].

Worse ambulation capacity prior to COVID-19 (OR: 11.3) was also a predictor of reduced function at discharge. Our results corroborate a study that followed the functional trajectory of older adults among the people who had a mild to moderate disability before admission to the intensive care unit. In this study, 39.5% developed a severe disability, and of those who had a severe disability prior to their stay in the intensive care unit, one-third had an intra-hospital death [32,37]. 

In our study, it was not possible to verify the treatment intensities that each group received; however, Stripari et al. [38] analyzed the functional recovery of three groups of critically ill COVID-19 survivors following their hospital stay, with the application of the early mobilization protocol. Their results show that functionally independent patients who preserved their functional status during hospitalization recovered functionality (44%); other patients showed dependence at ICU discharge, but recovered their independence by hospital discharge (33.2%); and some functionally dependent patients who were dependent at ICU discharge had not recovered their functional status at hospital discharge (22.8%). This suggests that age and time spent on patient mobilization out of bed were independent factors associated with becoming physically dependent after their ICU stay.

Our results showed that older adults hospitalized during the 2nd wave suffered a greater worsening in ambulation capacity than those hospitalized during the 1st wave (OR: 4.8). We speculate that this finding could be related to the fact that there was a greater number of dependent and non-functional ambulators requiring hospitalization in this period, because of the 1st wave. According to a study conducted by Moura et al. [39], the analysis of the 1st and 2nd waves in Brazil showed that there was a swift increase in the rate of cases and deaths in the 2nd wave, and despite social distancing measures being required, they were not respected. Inevitably, the contamination rate was higher, which may explain the occurrence of more individuals with worse ambulation capacity prior to the 2nd wave of COVID-19.

Therefore, older adults with a lower ambulation capacity prior to COVID-19 had a worse ambulation prognosis at hospital discharge, as well as a higher prevalence of death. A longer hospitalization period can cause functional losses, especially in older adults with comorbidities and the need for sedation [19,40]; another study showed that frailty and age over 80 years were the main factors associated with functional decline after hospital discharge, due to chronic obstructive pulmonary disease [3].

According to Stam et al. [41], post-intensive care effects could be the next public health crisis to occur, as at least 20% of patients with COVID-19 require supportive care in intensive care units, and approximately 50% of all patients at different ages tend to develop post-intensive care syndrome.

It is known that older adults have exacerbated responses to SARS-CoV-2 infection, including muscle damage related to the intensification of myokine (muscle cytokine) production, resulting in an increased viral invasion of the peripheral and musculoskeletal nervous system and a heightened muscle inflammatory process, which causes symptoms such as fatigue, weakness, muscle atrophy, and myalgia [42]. Individuals hospitalized due to COVID-19 who were in critical condition are likely to be at greater risk of developing post-COVID syndrome [39], with the persistence of several signs and symptoms—such as chronic pain—that can further affect the recovery process and, thus, require a longer rehabilitation time [22,27].

In our opinion, direct effects from COVID-19 infection and indirect effects due to hospitalization and drug treatment, mainly corticosteroids, can negatively impact muscle mass and strength, which would explain the impact on ambulation capacity. Most severe or moderate patients received corticoids and other drugs during hospitalization, which can potentially negatively impact muscle mass and strength. Both hospitalization and the related negative effects associated with medication, sedation, mechanical ventilation, and other interventions, plus the direct and indirect effects of SARS-CoV-2–such as local tissue lesions and secondary damage related to hyper-inflammatory responses–can be associated with functional deficiencies acquired during hospitalization. In fact, post-ICU syndrome associated with SARS-CoV-2 infection could contribute to the explanation of the impact on ambulation capacity [39,42,43].

Our results reinforce the importance of performing functional assessments and intervention for older adults during hospitalization, especially with regard to the ability to walk. The literature shows that gait speed is an indicator of future adverse health events in the older adult [27]. In addition, hospitalization itself is considered an aggravating factor for functionality, with a negative impact on mobility and the ability to perform daily activities [44]. 

One limitation of this study was the different methods used to assess the ambulation capacity of patients before and during hospitalization. Unfortunately, it was not possible to assess their ambulation capacity before COVID-19 through an evaluation done by physiotherapist, for obvious reasons. However, the participants included in our study presented appropriate cognitive capacity to recall their ambulation capacity before hospitalization.

However, another limitation of our study was that, despite all hospitalized patients receiving physiotherapy since their first day of hospitalization, unfortunately, we did not collect data regarding the number of sessions of physiotherapy for each patient.

Due to the high number of patients hospitalized during the pandemic and the lack of health professionals available at that time, unfortunately, it was not possible to screen important risk factors for ambulation impairment, due to factors such as the presence of frailty and sarcopenia in all patients. In fact, this screening was done only in the geriatric unit of our hospital, and it was not possible to use this data to compare with other general units.

Finally, only a few studies with a small number of participants have evaluated the impairment of ambulation in adults and older adults hospitalized with COVID-19 [32,33,45]. Therefore, post-hospital follow-up is necessary to investigate possible people suffering from Post COVID Syndrome [46,47]. Our study, despite the fact that it can only show associations, contributes to filling this gap in the literature and seeks to elucidate the functional impact of COVID-19 on older adults, helping to devise new interventions for COVID-19 prevention and rehabilitation related to this population.

## 5. Conclusions

The older patients who died during hospitalization due to COVID-19 had greater gait impairment prior to hospitalization, and more than half of those who were discharged had worsened gait impairment. The main factors associated with impaired ambulation after hospitalization were a hospital stay longer than 20 days; having an impaired gait prior to admission; and being infected by the 2nd wave of COVID-19. Our results suggest the need for interventions aiming to reduce ambulation impairment in hospitalized older people with severe COVID-19, in order to decrease the risk of longer-term functional impairment in this population.

## Figures and Tables

**Table 1 ijerph-20-07057-t001:** Sociodemographic and clinical characteristics of the older people according to their functionality during the pre-admission assessment.

Sociodemographic and Clinical Characteristics		Ability to Walk before Hospital Admission				
		Independent Ambulator *n* = 289 (67.4%)	Dependent Ambulator *n* = 68 (15.8%)	Non-Functional Ambulator *n* = 72 (16.8%)	Total *n* = 429 (100%)	*p*(*χ^2^*)
**Age (years)**	60–69	172 (69.9)	32 (13.0)	42 (17.1)	246	* **0.046** *
	70–79	95 (68.8)	24 (17.4)	19 (13.8)	138	
	80 or older	22 (48.9)	12 (26.7)	11 (24.4)	45	
**Sex**	Female	113 (66.1)	27 (15.8)	31 (18.1)	171	*0.828*
	Male	176 (66.2)	41 (15.9)	41 (15.9)	258	
**Race**	White	91 (52.3)	20 (13.7)	35 (24.0)	146	* **0.015** *
	Brown	88 (64.7)	26 (19.1)	22 (16.2)	139	
	Black	108 (75.5)	21 (14.7)	14 (9.8)	143	
**Habits**	Tobacco	62 (67.5)	12 (15.2)	5 (6.3)	79	* **0.016** *
	Alcoholic Beverage	8 (72.7)	2 (18.2)	1 (9.1)	11	*0.782*
**Comorbidity**	Immunosuppression	30 (69.8)	9 (20.9)	4 (9.3)	43	*0.299*
	Hematologic	3 (75.0)	1 (25.0)	-	4	*0.629*
	Neurologic	10 (41.7)	8 (33.3)	6 (25.0)	24	* **0.015** *
	Pulmonary	20 (65.3)	5 (17.9)	5 (16.8)	30	*0.992*
	Cardiovascular	62 (63.26)	17 (17.35)	16 (19.39)	95	*0.818*
	Renal	8 (80.0)	2 (20.0)	-	10	*0.355*
	Hepatic	4 (80.0)	1 (20.0)	-	5	*0.599*
	Systemic Arterial Hypertension	153 (67.7)	40 (17.7)	33 (14.6)	226	*0.302*
	Diabetes Mellitus	111 (70.3)	29 (18.3)	18 (11.4)	158	*0.060*
	Obesity	49 (68.1)	15 (20.8)	8 (11.1)	72	*0.221*
	Dyslipidemia	37 (77.1)	6 (12.5)	5 (10.4)	48	*0.190*
**Hospitalization (>20 days)**		118 (66.3)	36 (22.2)	24 (13.5)	178	*0.059*
**Intensive care unit (>11 days)**		135 (65.2)	38 (18.4)	34 (16.4)	207	*0.332*
**Invasive mechanical ventilation**		265 (65.59)	57 (14.11)	82 (20.30)	404	* **0.001** *
**In-hospital death**		106 (64.17)	31 (13.37)	50 (22.46)	187	*0.684* * **˂0.001** *
**COVID-19 wave**	1st	107 (59.1)	22 (12.2)	52 (28.7)	181	
	2nd	182 (73.4)	46 (18.5)	20 (8.1)	248	* **˂0.001** *

Abbreviation: ^χ2^ = *chi square. In bold, p-values that present a statistical difference.*

**Table 2 ijerph-20-07057-t002:** Ambulation capacity before COVID-19, in-hospital death, and reduced ambulation capacity at hospital discharge.

Ability to Walk before Hospitalization	In-Hospital Death *n* = 187 (43.6%)	At Hospital Discharge *n* = 252 (57.4%)		Total	
		Same Ambulation Capacity *n* = 102 (23.81%)	Worse Ambulation Capacity *n* = 140 (32.6%)	*n* = 429	*p*
**Independent Ambulator *(n* = 289)**	106 (24.7)	78 (18.2)	105 (24.5)	289 (67.4)	***<0.001*** ^**χ^2^**^
**Dependent Ambulator (*n* = 68)**	31 (7.2)	2 (0.5)	35 (8.2)	68 (15.9)	
**Non-Functional Ambulator (*n* = 72)**	50 (11.7)	22 (5.1)	--	72 (16.8)	

Abbreviation: ^χ^2^^ = *chi square. In bold, p-values that present a statistical difference*.

**Table 3 ijerph-20-07057-t003:** Distribution and bivariate regression for the ambulation capacity change at hospital discharge (*n* = 242).

Demographic and Clinical Characteristics		Ambulation Capacity at Hospital Discharge			
		Same *n* = 102 (42.1%)	Worse *n* = 140 (57.9%)	OR (CI95%)	*p*
**Age (years)**	60–69	61 (42.9)	81 (57.1)	Ref	
	70–79	35 (44.3)	44 (55.7)	1.0 (0.5–1.6)	*0.847*
	80 years or older	6 (28.6)	15 (71.4)	1.8 (0.7–5.1)	*0.217*
**Sex**	Female	57 (50.0)	57 (50.0)	Ref	
	Male	45 (35.2)	83 (64.8)	**1.8 (1.1–3.1)**	* **0.020** *
**Color/Race**	White	47 (58.8)	33 (41.2)	Ref	
	Brown	27 (33.3)	54 (66.7)	**2.8 (1.5–5.4)**	* **0.001** *
	Black	27 (34.6)	51 (65.4)	**2.7 (1.4–5.1)**	* **0.003** *
**Habits**	Tobacco	13 (30.9)	29 (69.1)	**1.7 (0.8–3.5)**	* **0.124** *
	Alcoholic beverage	–	4 (100.0)	---	*---*
**Comorbidities**	Immunosuppression	9 (36.0)	16 (64.0)	1.3 (0.6–3.1)	*0.512*
	Hematologic	1 (50.0)	1 (50.0)	0.7 (0.1–11.7)	*0.822*
	Neurologic	3 (721.4)	11 (78.6)	**2.8 (0.7–10.3)**	* **0.120** *
	Pulmonary	9 (56.3)	7 (43.7)	0.5 (0.2–1.5)	*0.243*
	Cardiovascular	29 (46.0)	34 (54.0)	0.8 (0.4–1.4)	*0.468*
	Renal	2 (33.3)	4 (66.7)	1.5 (0.3–8.2)	*0.660*
	Hepatic	1 (33.3)	2 (66.7)	1.4 (0.1–16.3)	*0.757*
	Systemic Arterial Hypertension	62 (44.9)	76 (55.1)	0.7 (0.4–1.3)	*0.314*
	Diabetes Mellitus	42 (42.4)	57 (57.6)	1.0 (0.6–1.6)	*0.942*
	Obesity	12 (30.8)	27 (69.2)	**1.8 (0.8–3.7)**	* **0.119** *
	Dyslipidemia	4 (19.1)	17 (80.9)	**3.4 (1.1–10.4)**	* **0.033** *
**Hospitalization (>20 days)**		32 (28.3)	81 (71.7)	**2.9 (1.7–5.0)**	* **<0.001** *
**Intensive care unit (>11 days)**		29 (19.0)	71 (71.0)	**1.9 (1.1–3.4)**	* **0.027** *
**Invasive mechanical ventilation**		91 (43.1)	120 (56.9)	0.6 (0.3–1.5)	*0.211*
**Ambulation capacity before COVID-19**	Independent Ambulator	78 (42.6)	105 (57.4)	Ref	
	Dependent Ambulator	2 (5.4)	35 (94.6)	**13.0 (3.0–55.6)**	* **0.001** *
**2nd wave**		28 (26.4)	106 (73.6)	**5.2 (3.0–9.1)**	* **<0.001** *

Abbreviations: Ref = reference; OR = odds ratio; CI = 95% confidence interval; *p* = probability of significance. In bold, odds ratio (OR) and *p*-values that present a statistical difference.

**Table 4 ijerph-20-07057-t004:** Logistic regression adjusted for the ambulation capacity evolution at hospital discharge (*n* = 214).

Demographic and Clinical Characteristics		Ambulation Capacity at Hospital Discharge	
		OR (CI_95%_)	*p*
Sex	Male	1.7 (0.9–3.6)	*0.114*
Color/Race	White	Ref	
	Brown	1.7 (0.7–4.4)	*0.203*
	Black	1.1 (0.4–2.7)	*0.925*
Tobacco Use		0.7 (0.3–1.9)	*0.603*
Obesity		1.2 (0.4–3.4)	*0.770*
Dyslipidemia		1.3 (0.3–5.6)	*0.741*
Neurological Diseases		3.8 (0.3–46.9)	*0.293*
**Hospitalization > 20 days**		**3.5 (1.7–7.3)**	* **0.001** *
**Ambulation capacity before hospital admission**	Independent Ambulator	Ref	
	Dependent Ambulator	**11.3** **(1.4–52.7)**	* **0.002** *
**2nd wave**		**4.8** **(2.1–11.1)**	* **<0.001** *

Abbreviations: OR = odds ratio; CI = 95% confidence interval. In bold, odds ratio (OR) and *p*-values that present a statistical difference

## Data Availability

Where data is unavailable due to privacy or ethical restrictions.

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
