# Peer review of "Risk Factors for the Impairment of Ambulation in Older People Hospitalized with COVID-19: A Retrospective Cohort Study"

_ijerph, 2023, doi:10.3390/ijerph20227057_

Round 1
Reviewer 1 Report
Comments and Suggestions for Authors
The authors conducted a retrospective analysis referring to older (60+) patients hospitalized with Covid -19. The aim of the study was to find factors associated with impairment of walking ability at discharge.
Methods: the ability to walk before hospital admission was based on patients report. Patients were then divided into three groups depending on their ability to walk.
Ambulation at discharge was compared to the ability to walk before hospital admission. Factors associated with impairment of ambulation were than identified and regression analyses was conducted.
Methods: the ability to walk before hospital admission based on patient reports and compared to the reports documented by the physiotherapists at discharge. So different measures were compared. This should be discussed in more detail. Furthermore, the number of unites of physiotherapy administered to the patients should also be mentioned since this may have an impact on the outcome. In addition, how were patients treated for covid and how long? Did they receive corticoids for instance, a drug known for it´s negative impact on muscles. Did the groups differ referring to these issues?
How many patients with impaired ability to walk at admission improved during hospital stay? It would expect, that some people improved despite covid-19 infection.
Were the patients assessed for frailty at admission? I think some of the patients were frail and frailty is a well known risk factor for any functional decline.
A retrospective analyses can only show associations. Please discuss this issue also.
In your conclusion you say that the results suggest that there is a need for intervention. However, how did you intervene during hospital stay anyway and was there a difference of intervention intensity between groups?
Did patients with impaired ambulation at admission receive more and earlier physiotherapy for instance?
Please explain the treatments carried out in more detail and look for differences of treatment intensity between groups.
There might be a dose response relationship between improvement and intensity of treatment in each group.
Reviewer 2 Report
Comments and Suggestions for Authors
This study addresses the prevalence of factors associated with reduced ambulatory capacity at hospital discharge in older people admitted with Covid-19.
Major concerns
The conclusion is that staying n hospital for more than 20 days results in reduced ambulatory capacity. Is there any research into the the effect of inactivity on muscle strength and mass that affects ambulatory capacity in any age group, particularly older individuals or even animal studies. The point should be made that the disease is the cause of reduced activity which leads to this loss in muscle mass and function. Please consider addressing this in the Introduction and Discussion sections.
Minor
Line 15: Change to "Some older people hospitalized with Covid-19 have experienced reduced ambulatory capacity."
Line 16: Change "need" to "needs."
Line 21: Change "his" to "their."
Line 23: The sentence beginning "Data from..." is incomplete.
Line 25: What is OR?
Line 26: Change to "...who survived following hospitalization due..."
Line 37: Delete "the"
Line 38: Change to "There are more than 771,154,224 confirmed cases of Covid-19 and more than 6,960,783 deaths due to the disease, worldwide ."
Line 44: The word "functional impairment" seems out of place. Please correct this sentence.
Line 47: Delete "the"
Line 72: Please add an explanation for RedCap. Also, delete "older"
Methodology and Results sections are clearly written.
Comments on the Quality of English Language
Please see language suggestions in my general comments.
Round 2
Reviewer 1 Report
Comments and Suggestions for Authors
all comments of the reviewer were taken into account